# Design, Synthesis, and Antitumor Activity of Olmutinib Derivatives Containing Acrylamide Moiety

**DOI:** 10.3390/molecules26103041

**Published:** 2021-05-20

**Authors:** Xiaohan Hu, Sheng Tang, Feiyi Yang, Pengwu Zheng, Shan Xu, Qingshan Pan, Wufu Zhu

**Affiliations:** School of Pharmacy, Jiangxi Science & Technology Normal University, Nanchang 330013, China; huxiaohan1717@163.com (X.H.); shengtang_1219@163.com (S.T.); 18380539689@163.com (F.Y.); Zhengpw@126.com (P.Z.); Shanxu9891@126.com (S.X.)

**Keywords:** acrylamide, olmutinib derivatives, EGFR, inhibitor

## Abstract

Two series of olmutinib derivatives containing an acrylamide moiety were designed and synthesized, and their IC_50_ values against cancer cell lines (A549, H1975, NCI-H460, LO2, and MCF-7) were evaluated. Most of the compounds exhibited moderate cytotoxic activity against the five cancer cell lines. The most promising compound, **H10**, showed not only excellent activity against EGFR kinase but also positive biological activity against PI3K kinase. The structure–activity relationship (SAR) suggested that the introduction of dimethylamine scaffolds with smaller spatial structures was more favorable for antitumor activity. Additionally, the substitution of different acrylamide side chains had different effects on the activity of compounds. Generally, compounds **H7** and **H10** were confirmed as promising antitumor agents.

## 1. Introduction

Recent studies have shown that the mutation rate and overexpression rate of epidermal growth factor receptor (EGFR; a transmembrane protein) of patients with non-small-cell lung cancer (NSCLC) are as high as 75% in vivo [1,2]. EGFR is associated with various growth processes of cancer, including proliferation, differentiation, migration, apoptosis, and angiogenesis [3]. Therefore, EGFR has garnered considerable attention as an antitumor drug target, giving rise to numerous investigations of EGFR kinase inhibitors to disrupt and inhibit the proliferation and growth of tumor cells [4]. The first-generation EGFR inhibitor gefitinib (**1**) (Figure 1) and the second-generation EGFR inhibitor afatinib (**2**) have good inhibitory effects on the EGFR^L858R^ mutation and the EGFR^T790M^ mutation, respectively [5,6,7]. However, the first- and second-generation inhibitors have poor kinase selectivity between the EGFR^T790M^ mutant and the wild type. Moreover, their clinical efficacy is limited. The third-generation EGFR inhibitor olmutinib [8,9] (**3**) was developed by Hanmei Pharmaceutical Company, and is an irreversible inhibitor for the treatment of patients with locally advanced or EGFR^T790M^-mutant NSCLC (Figure 2) [10]. The IC_50_ values of **3** against HCC827 (EGFR^dell19^), H1975 (EGFR^L858R/T790M^), and A549 (EGFR^WT^) cells were 9.2 nM, 10 nM, and 225 nM, respectively. This indicated that **3** had strong selectivity for the EGFR mutation. However, **3** had toxic side effects such as palmoplantar keratoderma and diarrhea. Therefore, the goal of this study to obtain new olmutinib derivatives which can overcome the side effects of skeleton migration.

To guide our modification, a molecular docking simulation of **3** and EGFR protein was performed. The thienopyrimidine structure of **3** was inserted into the larger hydrophobic band of the protein and could form hydrogen bonds with MET-793 in the hinged region, as shown in Figure 2A,B. Therefore, we transformed the thiophene pyrimidine structure into thiophene and pyrimidine to explore the antiproliferative activity of the target compounds when fully occupying the protein cavity. At the same time, inspired by WZ4002 (**4**), we replaced the phenylpiperazine structure of **3** with an anisidine side chain and an electron-withdrawing cyano group [11]. Different acrylamide moieties were used to explore the influence of side-chain length and halogen atoms on the activities of compounds. We used this design idea to synthesize the first series of target compounds (Figure 3).

To design of the second series of target compounds (Figure 4), we retained the thiophene and pyrimidine core and introduced the triazine structure of phosphatidylinositol-3-kinase (PI3K) inhibitor ZSTK474 (**5**) while at the same time, the structures of thiophene and pyrimidine were transformed into triazine and pyrimidine [12,13]. The introduction of the 1,3,5-triazine ring increased the polarity of the molecule to better form key hydrogen bonds with MET-793 [14]. The oxygen of the morpholine ring of GDC-0941 (**6**) can form hydrogen bonds with the VAL-851 of the PI3K hinge region. Moreover, the introduction of the morpholine ring enhanced the mTOR-inhibitory activity of the compound; thus, the morpholine ring group was preserved [15,16,17]. Michael receptors were introduced in the solvent region to explore the interaction with the surrounding amino acids. Based on this design strategy, we completed the synthesis of the second series of compounds. We expected to obtain an ideal inhibitor with a better EGFR-inhibitory activity and to explore whether the target compound has an inhibitory effect on PI3K following the modifications described above.

## 2. Results and Discussion

### 2.1. Chemistry

According to the structure-based drug design (SBDD) strategy, we designed and synthesized two series of olmutinib derivatives as EGFR inhibitors containing an acrylamide moiety. The synthetic routes of target compounds **H1–H16** are outlined in Scheme 1 and Scheme 2.

As shown in Scheme 1, we used commercially available 2,4,6-trichloropyrimidine (**7**) and thiophen-2-ylboronic acid to obtain **11a–11b** through cyclization, chlorination, and nucleophilic substitution reactions. Compounds **11a–11b** reacted with different amide side chains to give the target compounds **H1–H8**. As shown in Scheme 2, we used 1,3,5-triazine as the starting material to obtain **18a–18b** with different amines via six steps of substitution, reduction, and chlorination. Finally, **18a–18b** reacted with different amide side chains to give the target compounds **H9–H16**. The structural information of target compounds was confirmed by ^1^H-NMR, ^13^C-NMR, and TOF MS (ES+), the results of which were consistent with the structures depicted.

### 2.2. Biological Discussion

Four human tumor cell lines (A549, H1975, NCI-H460, and MCF-7) and human normal cell line LO2 were selected to evaluate the antiproliferative activity of all target compounds in vitro. Olmutinib was used as a positive control. Results are summarized in Table 1, where the values are the average of at least three independent experiments. Compared with the lead compound olmutinib, most target compounds were less toxic to the normal cell line LO2, which indicates that the target compounds had a selective inhibitory effect on cancer cells. After the introduction of electron-withdrawing cyanide, the antiproliferative activities of compounds **H1–H6** that were substituted with cyano groups showed moderate inhibitory activity against all the cell lines. However, the antiproliferative activities of the compounds that were substituted with the anisidine side chain performed better than the cyano group chain. This indicates that the introduction of the electron-withdrawing units could not increase the antiproliferative activity of the compounds. Obviously, compound **H7** showed the greatest inhibitory activities against A549 and H1975 cancer cell lines, with IC_50_ values of 4.37 ± 0.50 μM and 4.59 ± 0.46 μM, respectively, which were similar to the reference compound of olmutinib.

Table 1 shows that the introduction of the 1,3,5-triazine ring and morpholine ring significantly enhanced the antiproliferation activity of the compounds. At the same time, we found that the antiproliferative activities of the compounds that were substituted with dimethylamine groups, were better than those that were substituted with diethylamine groups. Therefore, we speculate that the inner cavity area of the hydrophobic region is limited and only can accommodate molecules with smaller structures. From the docking results (Figure 5B), it was found that the dimethylamine group of compound **H10** penetrated into the interior of the protein (4zau) and completely occupied the space, which suggests that the introduction of a larger group than dimethylamine will not increase the activity of these compounds. The optimal compound **H10** showed excellent antiproliferative activity against A549 and MCF-7 cancer cell lines, with IC_50_ values of 3.36 ± 1.59 μM and 13.05 ± 1.36 μM, respectively, which were superior to the drug of reference. The selectivity of compound **H10** to A549 cells was 29.76 times than to LO2 cells, and about 5 times that of the lead compound olmutinib.

Compounds **H7** and **H10** had excellent antiproliferative activity and were further screened out for kinase inhibition testing. We evaluated compounds **H7** and **H10** with EGFR^T790M/L858R^ kinases and PI3Kα kinases. As shown in Table 2, compounds **H7** and **H10** exhibited potent inhibition against EGFR^T790M/L858R^ kinase. In particular, compound **H10** showed better inhibitory activity against PI3Kα kinase than the control drug olmutinib. These data demonstrate that compound **H10** is expected to be a dual inhibitor of EGFR and PI3K.

### 2.3. Molecular Docking Study

To explore the binding mode of the target compound (**H10)** with the active site of EGFR, molecular docking simulation was carried out using AutoDock 4.2 software. The docking tutorials and detailed explanations of the AutoDock basic methods we used can be found at the following address: http://autodock.scripps.edu/faqs help/tutorial (accessed on 25 March 2021). According to the analysis results of the cells and kinases, we chose compound **H10** as the example ligand; the structures of EGFR^WT^ (PDB CODE: 4zau) and EGFR^T790M^ (PDB CODE: 3ika) were selected as docking models. 

The combination of compound **H10** with the EGFR^T790M^ and PI3Kγ molecular active sites is shown in Figure 5. When compound **H10** docked with the 3ika (Figure 5A), we observed that the dimethylamine group extended into the ATP hydrophobic pocket and formed hydrogen bonds with residues LYS-745, and the amino group formed hydrogen bonds with MET-793 residues. Figure 5C shows that the morpholine rings of compound **H10** formed hydrogen bonds with the VAL-882 residues in 3L08. Among these, the acrylamide side chain of compound **H10** formed hydrogen bonds with the LYS-833 residue in 3L08. This was in line with the combination model we predicted before. The abovementioned SAR (structure–activity relationship) analysis and molecular docking results indicate that compound **H10** could be a potentially interesting anticancer agent.

## 3. Experimental Section

### 3.1. General Information

Unless otherwise stated, all reagents used in the experiment were purchased at commercial analytical grade and used directly without further purification. Common solvents (ethanol, methanol, petroleum ether, ethyl acetate, dichloromethane 1,2-dimethoxyethane, etc.) were absolutely anhydrous. All reactions were monitored on a GF254 thin-layer chromatography plate (Laishan Penghan Plastic Industry Store, Yantai, Shandong, China), and spots were visualized at 254 nanometers or 365 nanometers with iodine or light. The target compound (20 mg) and 3.5 mL DMSO formed the corresponding compound solution, and the structure of the target compound was confirmed by ^1^H-NMR and ^13^C-NMR on a Bruker 400 MHz spectrometer (Bruker Bioscience, Billerica, MA, USA) using tetramethylsilane (TMS) as an internal standard at room temperature (see Appendix A). The target compounds (0.5 mg) and LC-MS methanol were prepared in a 0.5 µg/mL mixed solution, and mass spectrometry (MS) of target compounds was carried out on a Waters High Resolution Quadrupole Time of Flight Tandem Mass Spectrometer (Waters, Milford, MA, USA, Xevo G2-XS Tof). The purity of all compounds was determined using an Agilent 1260 liquid chromatograph equipped with an Inertex-C18 column. The purity of all target compounds was ≥95%.

### 3.2. Chemistry

#### 3.2.1. Representative Procedure for the Synthesis of 2,4-dichloro-6-(thiophen-2-yl)pyrimidine (**8**)

2,4,6-Trichloropyrimidine **7** (70.0 g, 381.6 mmol) and thiophen-2-ylboronic acid (25.2 g, 196.8 mmol) were coupled by coupling reaction in 1,2-dimethoxyethane and water 5:1 solvent. The above solution was stirred at 90 °C for about 1.5 h. The reaction was monitored by TLC. After the reaction, the reactant was purified via silica gel column chromatography to obtain compound **8**. Yield: 95%; color: yellow; m.p.: 145.1–147.5 °C. ^1^H NMR (400 MHz, DMSO-*d*_6_) *δ* 7.49 (s, 1H), 7.23 (s, 1H), 7.20 (s, 1H), 7.19 (s, 1H). TOF MS ES+ (*m/z*): [M + H]^+^, calcd for C_8_H_4_Cl_2_N_2_S: 232.1100, found, 232.1103.

#### 3.2.2. Representative Procedure for the Synthesis of 2-chloro-4-(3-nitrophenoxy)-6-(thiophen-2-yl)pyrimidine (**9**)

Intermediate **8** (19.0 g, 82.3 mmol), metanitrophenol (12.0 g, 86.3 mmol), and cesium carbonate (32.0 g, 99.5 mmol) were dissolved in 1,4-dioxane (120 mL), and then stirred for 4 h at room temperature. The completion of the reaction was monitored by TLC. After the reaction, the reaction solvent was removed under reduced pressure to obtain a yellow solid **9**. Yield: 91.4%; color: yellow; m.p.: 152–155 °C. ^1^H NMR (400 MHz, DMSO-*d*_6_) *δ* 8.05 (s, 1H), 8.02 (s, 1H), 7.64 (s, 1H), 7.54 (m, 1H), 7.24 (s, 1H), 7.21 (s, 1H), 7.19 (s, 1H), 7.16 (s, 1H). TOF MS ES+ (*m/z*): [M + H]^+^, calcd for C_14_H_8_ClN_3_O_3_S: 333.7869, found, 333.7872.

#### 3.2.3. Representative Procedure for the Synthesis of **10a–10b**

Intermediate **9** (5.3 g, 16.0 mmol), toluene-p-sulfonic acid (5.6 g, 32.5 mmol) and different amino side chains (16.0 mmol) were dissolved in acetonitrile (60 mL) and stirred for 4–5 h at 100 °C. After the reaction, the reaction solvent was removed under reduced pressure and a large amount of solid precipitated after adding twice the amount of water. The solid was filtered at atmospheric pressure and dried to obtain **10a–10b**.

N-(3-methoxyphenyl)-4-(3-nitrophenoxy)-6-(thiophen-2-yl)pyrimidin-2-amine (**10a**) 

Yield: 90.1%; color: yellow; m.p.: 169.5–172.7 °C. ^1^H NMR (400 MHz, DMSO-*d*_6_) *δ* 8.36 (s, 1H), 8.05 (s, 1H), 7.91 (s, 1H), 7.89 (s, 1H), 7.73 (s, 1H), 7.67 (s, 1H), 7.57–7.54 (m, 1H), 7.30 (dd, *J* = 7.5, 1.5 Hz, 1H), 7.24 (d, *J* = 7.4 Hz, 1H), 7.23–7.16 (m, 1H), 7.14 (dt, *J* = 7.5, 1.5 Hz, 1H), 6.89 (s, 1H), 6.57 (dt, *J* = 7.5, 1.6 Hz, 1H), 3.80 (s, 3H). TOF MS ES+ (*m/z*): [M + H]^+^, calcd for C_21_H_15_N_3_O_4_S: 405.0925, found, 405.0927.

3-((4-(3-nitrophenoxy)-6-(thiophen-2-yl)pyrimidin-2-yl)amino)benzonitrile (**10b**) 

Yield: 80.7%; color: yellow; m.p.: 176.1–179.3 °C. ^1^H NMR (400 MHz, DMSO-*d*_6_) *δ* 8.23 (s, 1H), 8.05 (dt, *J* = 7.5, 1.6 Hz, 1H), 7.94 (s, 1H), 7.84 (s, 1H), 7.64 (dd, *J* = 7.3, 1.6 Hz, 1H), 7.55 (t, *J* = 7.5 Hz, 1H), 7.45–7.40 (m, 1H), 7.39 (d, *J* = 7.5 Hz, 1H), 7.33 (dt, *J* = 7.1, 1.7 Hz, 1H), 7.24 (dd, *J* = 7.5, 1.6 Hz, 1H), 7.19 (t, *J* = 7.4 Hz, 1H), 7.14 (dt, *J* = 7.5, 1.5 Hz, 1H), 6.89 (s, 1H). TOF MS ES+ (*m/z*): [M + H]^+^, calcd for C_21_H_12_N_4_O_3_S: 401.0746, found, 401.0749.

#### 3.2.4. Representative Procedure for the Synthesis of **11a–11b**

Intermediate **10a–10b** (13mmol), ferric chloride (4.2 g, 15.6 mmol), and activated carbon (1.1 g, 91.0 mmol) were dissolved in ethanol. The solution was warmed to 80 °C, and then water and hydrazine were added (6.5 g, 130 mmol) and the solution stirred for 4 h. After the reaction, the reaction solvent was removed under reduced pressure, saturated sodium bicarbonate aqueous solution was added, and the solution was filtered to generate **11a–11b**.

4-(3-aminophenoxy)-N-(3-methoxyphenyl)-6-(thiophen-2-yl)pyrimidin-2-amine (**11a**) 

Yield: 78.3%; color: yellow; m.p.: 192.5–195.4 °C. ^1^H NMR (400 MHz, DMSO-*d*_6_) *δ* 8.46 (s, 1H), 7.83 (s, 1H), 7.74 (s, 1H), 7.66 (s, 1H), 7.27 (dt, *J* = 6.3, 4.5 Hz, 1H), 7.23 (dt, *J* = 6.8, 3.6 Hz, 1H), 7.22–7.20 (m, 1H), 7.17 (s, 1H), 7.02 (s, 1H), 6.91 (s, 1H), 6.83 (s, 1H), 6.57 (s, 1H), 6.17 (s, 1H), 5.06 (s, 2H), 3.80 (s, 3H). TOF MS ES+ (*m/z*): [M + H]^+^, calcd for C_21_H_17_N_3_O_2_S: 376.1405, found, 376.1408.

3-((4-(3-aminophenoxy)-6-(thiophen-2-yl)pyrimidin-2-yl)amino)benzonitrile (**11b**) 

Yield: 71.1%; color: yellow; m.p.: 189.7–193.5 °C. ^1^H NMR (400 MHz, DMSO-*d*_6_) *δ* 8.23 (s, 1H), 7.83 (s, 1H), 7.66 (s, 1H), 7.42 (dt, *J* = 10.2, 9.1Hz, 1H), 7.40 (dt, *J* = 8.7, 5.6 Hz, 1H), 7.33 (s, 1H), 7.25 (s, 1H), 7.21 (s, 1H), 7.19 (s, 1H), 7.02 (s, 1H), 6.92 (s, 1H), 6.83 (s, 1H), 6.17 (s, 1H), 5.06 (s, 2H). TOF MS ES+ (*m/z*): [M + H]^+^, calcd for C_21_H_14_N_4_OS: 371.0983, found, 371.0980.

#### 3.2.5. Representative Procedure for the Synthesis of Target Compounds **H1–H8**

Intermediate **11a–11b** (1.5 mmol) and bicarbonate (0.25 g, 3 mmol) were dissolved in dichloromethane and reacted with different amide side chains under ice-bath conditions. The completion of the reaction was monitored by TLC. The reactant was purified via silica gel column chromatography to obtain the target compounds **H1–H8** with high purity.

#### 3.2.6. Representative Procedure for the Synthesis of **12–16**

The specific operation was carried out according to our previous research and the physical data were in agreement with reported values [18].

#### 3.2.7. Representative Procedure for the Synthesis of **17a–17b**

Intermediate **16** (0.5 g, 1.5 mmol), different amino side chains (3 mmol), and two drops of N,N-diisopropylethylamine (DIPEA) were dissolved in isopropanol (50 mL) and stirred for 1 h at 75 °C. After the reaction, the reaction solvent was removed under reduced pressure to obtain a yellow liquid **17a–17b**.

1-(5-(4-chloro-6-morpholino-1,3,5-triazin-2-yl)thiophen-2-yl)-N,N-dimethylmethanamine (**17a**) 

Yield: 85.1%; color: yellow; m.p.: 174.2–175.4 °C. TOF MS ES+ (*m/z*): [M + H]^+^, calcd for C_14_H_18_ClN_5_OS: 339.9370, found, 339.9373.

N-((5-(4-chloro-6-morpholino-1,3,5-triazin-2-yl)thiophen-2-yl)methyl)-N-ethylethanamine (**17b**) 

Yield: 82.2%; color: yellow; m.p.: 163.8–165.1 °C. TOF MS ES+ (*m/z*): [M + H]^+^, calcd for C_16_H_22_ClN_5_OS: 368.2436, found, 368.2433.

#### 3.2.8. Representative Procedure for the Synthesis of **18a–18b**

Intermediate **17a–17b** (1 mmol), 3-aminophenol (0.1 g, 1.1 mmol), and tert-butoxide (0.22 g, 2 mmol) were dissolved in THF and then stirred for 1 h under ice-bath conditions. The reaction was monitored by TLC. After the reaction, the reaction solvent was removed under reduced pressure to obtain a yellow liquid **18a–18b**.

3-((4-(5-((dimethylamino)methyl)thiophen-2-yl)-6-morpholino-1,3,5-triazin-2-yl)oxy)aniline (**18a**) 

Yield: 78.2%; color: yellow; m.p.: 179.3–182.9 °C. TOF MS ES+ (*m/z*): [M + H]^+^, calcd for C_20_H_24_N_6_O_2_S: 413.5214, found, 413.5211.

3-((4-(5-((diethylamino)methyl)thiophen-2-yl)-6-morpholino-1,3,5-triazin-2-yl)oxy)aniline (**18b**) 

Yield: 85.1%; color: yellow; m.p.: 178.4–181.5 °C. TOF MS ES+ (*m/z*): [M + H]^+^, calcd for C_22_H_28_N_6_O_2_S: 441.6275, found, 441.6277.

#### 3.2.9. Representative Procedure for the Synthesis of Target Compounds **H9–H16**

The synthesis of target compounds **H9–H16** was similar to that of target compounds **H1–H8**.

 *N-(3-((2-((3-cyanophenyl)amino)-6-(thiophen-2-yl)pyrimidin-4-yl)oxy)phenyl)acrylamide* (**H1**). Yield: 40.3%; color: yellow; m.p.: 197.2–199.4 °C; ^1^H NMR (400 MHz, DMSO-*d*_6_) *δ* 10.34 (s, 1H), 9.95 (s, 1H), 8.07 (s, 1H), 7.85 (s, 2H), 7.70 (d, *J* = 6.3 Hz, 1H), 7.51 (d, *J* = 8.3 Hz, 1H), 7.46 (t, *J* = 8.1 Hz, 2H), 7.33 (s, 2H), 7.28–7.24 (m, 1H), 7.16 (s, 1H), 7.03–6.99 (m, 1H), 6.42 (dd, *J* = 16.9, 10.1 Hz, 1H), 6.27 (dd, *J* = 16.9, 2.0 Hz, 1H), 5.79–5.74 (m, 1H). TOF MS ES+ (*m/z*): [M + H]^+^, calcd for C_25_H_19_N_5_O_2_S: 456.3863, found, 456.3864. *(E)-N-(3-((2-((3-cyanophenyl)amino)-6-(thiophen-2-yl)pyrimidin-4-yl)oxy)phenyl)but-2-enamide* (**H2**). Yield: 58.3%; color: yellow; m.p.: 201.2–205.9 °C; ^1^H NMR (400 MHz, DMSO-*d**_6_*) *δ* 10.13 (s, 1H), 9.95 (s, 1H), 8.08 (d, *J* = 5.8 Hz, 2H), 7.85 (d, *J* = 5.1 Hz, 2H), 7.68 (d, *J* = 10.9 Hz, 1H), 7.47 (d, *J* = 8.2 Hz, 1H), 7.42 (d, *J* = 7.9 Hz, 1H), 7.34 (s, 2H), 7.27–7.24 (m, 1H), 7.16 (s, 1H), 6.99 (d, *J* = 8.1 Hz, 1H), 6.80 (dd, *J* = 15.3, 7.6 Hz, 1H), 6.14 (d, *J* = 15.3 Hz, 1H), 1.86 (d, *J* = 7.0 Hz, 3H). TOF MS ES+ (*m/z*): [M + H]^+^, calcd for C_25_H_19_N_5_O_2_S: 454.5743, found, 454.5746. *N-(3-((2-((3-cyanophenyl)amino)-6-(thiophen-2-yl)pyrimidin-4-yl)oxy)phenyl)-3-methylbut-2-enamide* (**H3**). Yield: 66.3%; color: yellow; m.p.: 215.5–218.7 °C; ^1^H NMR (400 MHz, DMSO-*d*_6_) *δ* 10.01 (s, 1H), 9.95 (s, 1H), 8.07 (d, *J* = 3.8 Hz, 1H), 7.85 (d, *J* = 4.9 Hz, 2H), 7.67 (s, 1H), 7.42-7.40 (m, 3H), 7.34 (s, 2H), 7.27–7.24 (m, 1H), 7.15 (s, 1H), 6.96 (d, *J* = 8.1 Hz, 1H), 5.86 (s, 1H), 2.12 (s, 3H), 1.85 (s, 3H). TOF MS ES+ (*m/z*): [M + H]^+^, calcd for C_25_H_19_N_5_O_2_S: 468.5745, found, 468.5747. *(E)-N-(3-((2-((3-cyanophenyl)amino)-6-(thiophen-2-yl)pyrimidin-4-yl)oxy)phenyl)hex-2-enamide* (**H4**). Yield: 48.6%; color: yellow; m.p.: 209.6–211.4 °C; ^1^H NMR (400 MHz, DMSO-*d*_6_) *δ* 10.15 (s, 1H), 9.95 (s, 1H), 8.07 (d, *J* = 3.7 Hz, 1H), 7.85 (d, *J* = 5.0 Hz, 2H), 7.68 (s, 1H), 7.48 (d, *J* = 8.4 Hz, 1H), 7.42 (t, *J* = 8.0 Hz, 2H), 7.34 (s, 2H), 7.26 (dd, *J* = 5.0, 3.8 Hz, 1H), 7.16 (s, 1H), 6.98 (d, *J* = 7.9 Hz, 1H), 6.82–6.76 (m, 1H), 6.10 (d, *J* = 15.4 Hz, 1H), 2.18 (d, *J* = 7.0 Hz, 2H), 1.46 (d, *J* = 7.3 Hz, 2H), 0.90 (d, *J* = 3.5 Hz, 3H). TOF MS ES+ (*m/z*): [M + H]^+^, calcd for C_25_H_19_N_5_O_2_S: 482.3266, found, 482.3265. *N-(3-((2-((3-cyanophenyl)amino)-6-(thiophen-2-yl)pyrimidin-4-yl)oxy)phenyl)-2-fluoroacrylamide* (**H5**). Yield: 33.1%; color: yellow; m.p.: 222.9–226.8 °C; ^1^H NMR (400 MHz, DMSO-*d*_6_) *δ* 10.46 (s, 1H), 9.97 (s, 1H), 8.08 (d, *J* = 4.0 Hz, 1H), 7.86 (d, *J* = 5.3 Hz, 1H), 7.70 (d, *J* = 10.7 Hz, 2H), 7.49 (t, *J* = 8.0 Hz, 2H), 7.34 (s, 2H), 7.27 (d, *J* = 4.7 Hz, 1H), 7.17 (s, 1H), 7.09 (d, *J* = 8.6 Hz, 1H), 5.78 (d, *J* = 3.7 Hz, 1H), 5.65 (s, 1H), 5.48–5.41 (m, 1H). TOF MS ES+ (*m/z*): [M + H]^+^, calcd for C_24_H_16_FN_5_O_2_S: 458.4987, found, 458.4989. *(E)-N-(3-((2-((3-cyanophenyl)amino)-6-(thiophen-2-yl)pyrimidin-4-yl)oxy)phenyl)-4-methylpent-2-enamide* (**H6**). Yield: 51.8%; color: yellow; m.p.: 223.1–225.4 °C ^1^H NMR (400 MHz, DMSO-*d*_6_) *δ* 10.19 (s, 1H), 9.94 (s, 1H), 8.06 (d, *J* = 3.8 Hz, 1H), 7.84 (d, *J* = 5.2 Hz, 2H), 7.68 (d, *J* = 4.5 Hz, 1H), 7.48 (d, *J* = 8.3 Hz, 1H), 7.42 (t, *J* = 8.1 Hz, 1H), 7.33 (s, 2H), 7.25 (t, *J* = 4.4 Hz, 1H), 7.15 (s, 1H), 6.98 (d, *J* = 8.0 Hz, 1H), 6.81 (d, *J* = 6.3 Hz, 1H), 6.79–6.74 (m, 1H), 6.06 (d, *J* = 15.4 Hz, 1H), 2.44 (dd, *J* = 13.1, 6.6 Hz, 1H), 1.03 (d, *J* = 6.5 Hz, 6H). TOF MS ES+ (*m/z*): [M + H]^+^, calcd for C_27_H_23_N_5_O_2_S: 482.5866, found, 482.5868. *(E)-N-(3-((2-((3-methoxyphenyl)amino)-6-(thiophen-2-yl)pyrimidin-4-yl)oxy)phenyl)but-2-enamide* (**H7**). Yield: 43.7%; color: yellow; m.p.: 231.7–234.3 °C; ^1^H NMR (400 MHz, DMSO-*d*_6_) *δ* 10.15 (s, 1H), 9.43 (s, 1H), 8.00 (s, 1H), 7.23 (s, 2H), 6.76 (d, *J* = 6.9 Hz, 5H), 6.08 (d, *J* = 15.2 Hz, 6H), 3.67 (s, 3H), 1.93 (d, *J* = 7.8 Hz, 3H). ^13^C NMR (101 MHz, DMSO-*d*_6_) *δ* 170.28, 163.61, 163.02, 159.31, 155.14, 154.13, 140.54, 140.28, 139.03, 133.31, 132.43, 130.17, 129.72, 128.50, 127.84, 126.10, 120.67, 116.37, 115.97, 113.36, 112.62, 55.11, 17.41. TOF MS ES+ (*m/z*): [M + H]^+^, calcd for C_25_H_22_N_4_O_3_S: 459.5356, found, 459.5359. *(E)-N-(3-((2-((3-methoxyphenyl)amino)-6-(thiophen-2-yl)pyrimidin-4-yl)oxy)phenyl)-4-methylpent-2-enamide* (**H8**). Yield: 22.6%; color: yellow; m.p.: 217.9–219.1 °C; ^1^H NMR 400 MHz, DMSO-*d*_6_) *δ* 9.83 (s, 2H), 7.55 (d, *J* = 8.8 Hz, 4H), 6.87 (d, *J* = 8.9 Hz, 5H), 6.75 (dd, *J* = 15.4, 6.4 Hz, 3H), 6.05 (s, 1H), 6.01 (s, 1H), 3.72 (s, 3H), 1.06 (s, 7H). TOF MS ES+ (*m/z*): [M + H]^+^, calcd for C_27_H_26_N_4_O_3_S: 487.6124, found, 487.6126. *N-(3-((4-(5-((dimethylamino)methyl)thiophen-2-yl)-6-morpholino-1,3,5-triazin-2-yl)oxy)phenyl)acrylamide* (**H9**). Yield: 76.8%; color: yellow; m.p.: 224.9–226.1 °C; ^1^H NMR (400 MHz, DMSO-*d**_6_*) *δ* 9.39 (s, 1H), 7.25 (s, 2H), 7.08 (s, 1H), 7.06 (s, 1H), 7.03 (s, 1H), 7.01 (s, 1H), 6.48–6.47 (m, 1H), 6.25–6.21 (m, 1H), 5.74 (s, 1H), 3.64 (s, 4H), 3.58 (s, 4H), 3.02 (s, 2H), 1.24 (s, 6H). TOF MS ES+ (*m/z*): [M + H]^+^, calcd for C_23_H_26_N_6_O_3_S: 466.5668, found, 466.5670. *(E)-N-(3-((4-(5-((dimethylamino)methyl)thiophen-2-yl)-6-morpholino-1,3,5-triazin-2-yl)oxy)phenyl)but-2-enamide* (**H10**). Yield: 63.5%; color: yellow; m.p.: 202.4–203.7 °C; ^1^H NMR ((400 MHz, DMSO-*d*_6_) *δ* 9.65 (s, 1H), 9.38 (s, 2H), 7.25 (s, 1H), 7.05 (s, 2H), 6.46 (s, 1H), 5.75 (s, 1H), 5.48 (s, 1H), 3.64 (s, 4H), 3.58 (s, 4H), 3.02 (s, 2H), 1.93 (s, 6H), 1.23 (s, 3H). ^13^C NMR (101 MHz, DMSO-*d*_6_) δ 170.01, 169.89, 168.95, 163.76, 163.12, 157.33, 157.13, 146.17, 142.55, 142.09, 136.74, 136.45, 134.96, 128.48, 124.20, 121.58, 117.67, 114.48, 113.04, 58.14, 36.37, 35.08, 32.23, 24.75. TOF MS ES+ (*m/z*): [M + H]^+^, calcd for C_24_H_28_N_6_O_3_S: 480.5873, found, 480.5875. *N-(3-((4-(5-((dimethylamino)methyl)thiophen-2-yl)-6-morpholino-1,3,5-triazin-2-yl)oxy)phenyl)-3-methylbut-2-enamide* (**H11**). Yield: 84.5%; color: yellow; m.p.: 198.7–199.8 °C; ^1^H NMR (400 MHz, DMSO-*d*_6_) *δ* 9.30 (s, 1H), 7.76 (s, 1H), 7.23 (s, 1H), 6.98 (s, 2H), 6.77 (s, 1H), 6.42 (s, 1H), 5.99 (s, 1H), 3.08-3.15 (m, 10H), 2.21 (s, 9H), 1.84 (s, 3H). TOF MS ES+ (*m/z*): [M + H]^+^, calcd for C_24_H_28_N_6_O_3_S: 482.6257, found, 482.6256. *(E)-N-(3-((4-(5-((dimethylamino)methyl)thiophen-2-yl)-6-morpholino-1,3,5-triazin-2-yl)oxy)phenyl)-4-methylpent-2-enamide* (**H12**). Yield: 78.5%; color: yellow; m.p.: 199.7–201.3 °C; ^1^H NMR (400 MHz, DMSO-*d*_6_) *δ* 9.85 (s, 1H), 9.39 (s, 1H), 7.76 (s, 1H), 7.28 (s, 1H), 7.07 (m, 2H), 6.83 (m, 1H), 6.46 (s, 1H), 6.10 (m, 1H), 3.64 (s, 4H), 3.58 (s, 4H), 3.02 (s, 2H), 1.93(s, 7H), 1.06 (s, 6H). TOF MS ES+ (*m/z*): [M + H]^+^, calcd for C_26_H_32_N_6_O_3_S: 508.6263, found, 508.6266. *N-(3-((4-(5-((diethylamino)methyl)thiophen-2-yl)-6-morpholino-1,3,5-triazin-2-yl)oxy)phenyl)methacrylamide* (**H13**). Yield: 64.7%; color: yellow; m.p.: 240.7–243.5 °C; ^1^H NMR (400 MHz, DMSO-*d*_6_) *δ* 9.71 (s, 1H), 7.78 (s, 1H), 7.35 (s, 1H), 6.97 (s, 1H), 6.54 (s, 1H), 6.02 (s, 1H), 5.82 (s, 1H), 5.60 (s, 1H), 5.51 (s, 1H), 3.75 (s, 6H), 1.98 (s, 4H), 1.89 (s, 4H), 1.16 (d, 3H), 1.00 (m, 6H). TOF MS ES+ (*m/z*): [M + H]^+^, calcd for C_26_H_32_N_6_O_3_S: 509.6956, found, 509.6953. *(E)-N-(3-((4-(5-((diethylamino)methyl)thiophen-2-yl)-6-morpholino-1,3,5-triazin-2-yl)oxy)phenyl)but-2-enamide* (**H14**). Yield: 35.8%; color: yellow; m.p.: 189.7–192.1 °C; ^1^H NMR (400 MHz, DMSO-*d*_6_) *δ* 9.84 (s, 1H), 7.74 (s, 1H), 7.29 (s, 1H), 6.96 (s, 1H), 6.78 (d, *J* = 6.9 Hz, 1H), 6.15 (s, 1H), 6.11 (s, 1H), 5.82 (s, 1H), 5.78 (s, 1H), 3.74 (s, 6H), 1.83 (m, 4H), 1.14 (d, 4H), 0.98 (m, 9H). TOF MS ES+ (*m/z*): [M + H]^+^, calcd for C_26_H_32_N_6_O_3_S: 509.6933, found, 509.6931. *N-(3-((4-(5-((diethylamino)methyl)thiophen-2-yl)-6-morpholino-1,3,5-triazin-2-yl)oxy)phenyl)-3-methylbut-2-enamide* (**H15**). Yield: 41.5%; color: yellow; m.p.: 215.2–217.1 °C; ^1^H NMR (400 MHz, DMSO-*d*_6_) *δ* 9.71 (s, 1H), 7.27 (s, 1H), 7.06 (s, 1H), 7.00 (s, 1H), 6.45 (s, 1H), 6.02 (s, 1H), 5.87 (s, 1H), 5.63 (s, 1H), 3.74 (s, 6H), 2.15 (s, 4H), 1.86 (s, 4H), 1.14 (s, 6H), 0.99 (s, 6H). TOF MS ES+ (*m/z*): [M + H]^+^, calcd for C_26_H_32_N_6_O_3_S: 523.8974, found, 523.8971. *(E)-N-(3-((4-(5-((diethylamino)methyl)thiophen-2-yl)-6-morpholino-1,3,5-triazin-2-yl)oxy)phenyl)-4-methylpent-2-enamide* (**H16**). Yield: 26.4%; color: yellow; m.p.: 222.9.2–226.4 °C; ^1^H NMR (400 MHz, DMSO-*d*_6_) *δ* 9.88 (s, 1H), 7.75 (s, 1H), 7.30 (s, 1H), 7.05 (s, 1H), 6.79 (s, 1H), 6.48 (s, 1H), 6.10 (s, 1H), 6.07 (s, 1H), 3.72 (s, 6H), 1.14 (s, 10H), 1.05 (d, *J* = 6.7 Hz, 12H). TOF MS ES+ (*m/z*): [M + H]^+^, calcd for C_26_H_32_N_6_O_3_S: 536.6475, found, 536.6473.

### 3.3. EGFR and PI3Kα Kinase Assay

The potent compounds **H7** and **H10** were tested for their activities against EGFR^T790M/L858R^ and PI3Kα enzyme using the Kinase-Glo Luminescent Kinase Assay, with olmutinib as a positive control. The specific operation was carried out according to our previous research [19,20].

### 3.4. Cytotoxicity Assay In Vitro

The in vitro cytotoxic activities of all compounds **H1–H16** were evaluated with A549, H1975, LO2, and MCF-7 cell lines using the standard MTT assay, with olmutinib as a positive control [21].

### 3.5. Docking Studies

The three-dimensional structure of EGFR (PDB code: 4azu, 3L08) was obtained from the RCSB Protein Data Bank. We used AutoDock 4.2 software (The Scripps Research Institute, USA) for molecular docking. The docking process mainly included fixing the exact residues, adding hydrogen atoms, removing irrelevant water molecules, adding charges, etc. All the docking results were processed and modified in Open-Source PyMOL 1.8. x software (https://pymol.org (accessed on 25 March 2021).).

## 4. Conclusions

In summary, two series of olmutinib derivatives containing an acrylamide moiety (**H1–H16**) were synthesized and the pharmacological results indicate that most of the compounds exhibited moderate cytotoxic activity against five cell lines (A549, H1975, NCI-H460, LO2, and MCF-7). Among the most effective compounds, the IC_50_ values of **H10** against A549 and H1975 cells were 3.36 ± 1.59 µM and 1.16 ± 1.53 µM, respectively. The structure–activity relationship (SAR) indicated that the introduction of small-molecule swelling of dimethylamine was more favorable for the activity of the compounds. The amino group and morpholine rings formed hydrogen bonds on the 3ika and 3l08 MET-793 residues, respectively. The kinase activity of compound **H10** on PI3K was higher than that of the lead compound olmutinib at 1 µM, which indicates that compound **H10** may be a new dual inhibitor of EGFR and PI3K. Further research will be conducted in the near future.

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
