# Peer review of "Design, Synthesis, and Antitumor Activity of Olmutinib Derivatives Containing Acrylamide Moiety"

_molecules, 2021, doi:10.3390/molecules26103041_

Round 1

Reviewer 1 Report

The peer-reviewed article presents new data on the synthesis of the Olmutinib derivatives containing acrylamide moiety with moderate cytotoxic activity against the four cancer cell lines. The results of synthesis, cytotoxicity studies and molecular docking are presented. Although it was not possible to establish a higher cytotoxicity for any of the synthesized new derivatives than for the parent Olmutinib, the results obtained may be of interest to the developers of this series of drugs and this mechanism of action for establishing important structure-activity relationships. The work as a whole can be recommended for publication, but there are some comments and suggestions.
1. In Schemes 1 and 2, in most cases, only the solvent and temperature are indicated under the reaction conditions, and important information about the second used reagent is not indicated (for example, in Scheme 1, the first reaction: the source of the thiophene fragment must be indicated, and so on).
2. The selectivity of the action of anticancer drugs in relation to non-cancerous cell lines is an important factor showing the prospects of their use. It would be advisable to study the anti-proliferative activity of the obtained compounds on at least one normal cell line and calculate the selectivity indices for tumor and non-tumor cells.
3. For all obtained compounds, in addition to NMR spectroscopy data, it is necessary to provide the data of elemental analysis or high-resolution mass spectrometry to confirm their composition, and data confirming the purity of the obtained compounds (for example, chromatography-mass spectrometry). According to the images given in the Supp;ementary information, some compounds (for example, H1, H2, H3, etc.) may contain impurities, which casts doubt on the results of their biological tests.

Author Response

Reviewer #1: The peer-reviewed article presents new data on the synthesis of the Olmutinib derivatives containing acrylamide moiety with moderate cytotoxic activity against the four cancer cell lines. The results of synthesis, cytotoxicity studies and molecular docking are presented. Although it was not possible to establish a higher cytotoxicity for any of the synthesized new derivatives than for the parent Olmutinib, the results obtained may be of interest to the developers of this series of drugs and this mechanism of action for establishing important structure-activity relationships. The work as a whole can be recommended for publication, but there are some comments and suggestions.

  1. In Schemes 1 and 2, in most cases, only the solvent and temperature are indicated under the reaction conditions, and important information about the second used reagent is not indicated (for example, in Scheme 1, the first reaction: the source of the thiophene fragment must be indicated, and so on).

Response: Thank you for your attention and careful review of our manuscript. According to your suggestion, we have modified the “reagents and conditions” in schemes 1 and 2, and arranged in order of reaction reagents, reaction solvents, reaction conditions and reaction time. Lines 82-85 and 87-91 in the manuscript have been modified in red font.

  1. The selectivity of the action of anticancer drugs in relation to non-cancerous cell lines is an important factor showing the prospects of their use. It would be advisable to study the anti-proliferative activity of the obtained compounds on at least one normal cell line and calculate the selectivity indices for tumor and non-tumor cells.

Response: Thanks to the reviewer’s advice and it is useful to improve our paper. We would like to reply to your comments with the following points:

  • We first screened all the target compounds for normal cell LO2 according to your opinion. Then, we selected normal cell line LO2 and lung cancer cell line A549 as comparison and the selectivity indices for tumor and non-tumor cells was calculated, the result has been added in the corresponding position in Table 1.
  • We analyzed the screening results and selectivity index of LO2 cells. The corresponding results were as follows: "Compared with the lead compound Olmutinib, most target compounds were less toxic to normal cell LO2, which indicated that the target compounds had a selective inhibitory effect on cancer cells." and "The selectivity of compound H10 to A549 cells was 29.76 times than that of LO2 cells, and about 5 times than that of the lead compound Olmutinib." Lines 102-107 and 129-130 in the manuscript have been modified in red font.
  • “LO2” have been added in the corresponding position in the full text. Lines 11, 307 and 319 in the manuscript have been modified in red font.
  1. For all obtained compounds, in addition to NMR spectroscopy data, it is necessary to provide the data of elemental analysis or high-resolution mass spectrometry to confirm their composition, and data confirming the purity of the obtained compounds (for example, chromatography-mass spectrometry). According to the images given in the Suppementary information, some compounds (for example, H1, H2, H3, etc.) may contain impurities, which casts doubt on the results of their biological tests.

Response: Thank you very much for reading our manuscript carefully. According to your suggestion, we first confirmed the composition and purity of the target compound by TOF MS (ES+). For hydrogen spectra that may contain impurities, we chose to reconfirm the structure information of the target compound through 1H-NMR and made a check., and the new spectra information has been uploaded in the supplementary material. Meanwhile, we performed 13C-NMR on the H7 and H10 of the two optimized compounds and confirmed the structural information. Because some compounds have poor solubility, 1H-NMR spectra can't be tested again, so they are not provided. The remaining spectrograms have been uploaded into the supplementary material. The newly added 13C-NMR and TOF MS (ES+) data are also added in the corresponding position in the manuscript and marked in red.

Reviewer 2 Report

The manuscript molecules-1177035 "Design, Synthesis and Antitumor Activity of Olmutinib Derivatives Containing Acrylamide Moiety" by Hu et.al. describes the synthesis of two series of Olmutinib derivatives containing acrylamide moiety and the study their anti-cancer activity. Two leader compounds were tested for activity against the kinases EGFR and PI3K.

Comments and remarks:

1) The authors should strengthen the Introduction part. It looks very short now. The topic of synthesis and problems in the scientific field has not been disclosed.

2) Since almost all compounds were obtained for the first time, new compounds should be characterized by few physical methods (1H, 13C NMR, IR spectroscopy and mass spectrometry). Images of all spectra should be added in supplementary materials.

3) The authors did not indicate the conditions (for example, concentration) of NMR experiments in the experimental part and supplementary materials.

4) The study does not assess the impact of test compounds on normal cells.

Author Response

Reviewer #2: The manuscript molecules-1177035 "Design, Synthesis and Antitumor Activity of Olmutinib Derivatives Containing Acrylamide Moiety" by Hu et.al. describes the synthesis of two series of Olmutinib derivatives containing acrylamide moiety and the study their anti-cancer activity. Two leader compounds were tested for activity against the kinases EGFR and PI3K.

  1. The authors should strengthen the Introduction part. It looks very short now. The topic of synthesis and problems in the scientific field has not been disclosed.

Response: Thank you very much for your kindly reminding. According to your suggestion, we added introduction to Olmutinib. As a third-generation EGFR inhibitor, Olmutinib was an irreversible inhibitor for the treatment of patients with locally advanced or EGFRT790M mutant NSCLC. By in-depth understanding of the side effects of Olmutinib toxicity, the theme of this article was elicited. Through skeleton transition, new Olmutinib derivatives were obtained to overcome Olmutinib side effects. Lines 33-40 in the manuscript have been modified in red font.

  1. Since almost all compounds were obtained for the first time, new compounds should be characterized by few physical methods (1H, 13C NMR, IR spectroscopy and mass spectrometry). Images of all spectra should be added in supplementary materials.

Response: Thank you very much for reading our manuscript carefully. According to your suggestion, we first confirmed the composition and purity of the target compound by TOF MS (ES+). For hydrogen spectra that may contain impurities, we chose to reconfirm the structure information of the target compound through 1H-NMR and made a check., and the new spectra information has been uploaded in the supplementary material. Meanwhile, we performed 13C-NMR on the H7 and H10 of the two optimized compounds and confirmed the structural information. Because some compounds have poor solubility, 1H-NMR spectra can't be tested again, so they are not provided. The remaining spectrograms have been uploaded into the supplementary material. The newly added 13C-NMR and TOF MS (ES+) data are also added in the corresponding position in the manuscript and marked in red.

The authors did not indicate the conditions (for example, concentration) of NMR experiments in the experimental part and supplementary materials.

Response: Thanks for your comments on our work. According to your suggestion, we added experimental conditions of NMR experiment. The added content is as follows: "The target compounds (20 mg) and 3.5 mL DMSO were formed corresponding compound solution, and the structure of the target compound was confirmed by 1H-NMR and 13C-NMR on a Bruker 400 MHz spectrometer (Bruker Bioscience, Billerica, MA, USA) using tetramethylsilane (TMS) as an internal standard at room temperature. The target compounds (0.5 mg) and LC-MS methanol were prepared into 0.5 µg/mL mixed solution, and Mass spectrometry (MS) of target compounds were carried out by Waters High Resolution Quadrupole Time of Flight Tandem Mass Spectrometry (Waters, Xevo G2-XS Tof)." Lines 172-179 in the manuscript have been modified in red font.

  1. The study does not assess the impact of test compounds on normal cells.

Response: Thanks to the reviewer’s advice and it is useful to improve our paper. I would like to reply to your comments with the following points:

  • We first screened all the target compounds for normal cell LO2 according to your opinion. Then, we selected normal cell line LO2 and lung cancer cell line A549 as comparison and the selectivity indices for tumor and non-tumor cells was calculated, the result has been added in the corresponding position in Table 1.
  • We analyzed the screening results and selectivity index of LO2 cells. The corresponding results were as follows: "Compared with the lead compound Olmutinib, most target compounds were less toxic to normal cell LO2, which indicated that the target compounds had a selective inhibitory effect on cancer cells." and "The selectivity of compound H10 to A549 cells was 29.76 times than that of LO2 cells, and about 5 times than that of the lead compound Olmutinib." Lines 102-107 and 129-130 in the manuscript have been modified in red font.
  • “LO2” have been added in the corresponding position in the full text. Lines 11, 307 and 319 in the manuscript have been modified in red font.

Round 2

Reviewer 1 Report

The authors took into account the comments and suggestions. The article can be recommended for publication in the present form.

Author Response

Thanks for. Our manuscript molecules-1177035 was revised according to the academic editor's comments (any additions or modifications to the manuscript are shown in red): The itemized response to each comment was attached.
Reviewers' comments: Major revision is require regard the Chemistry.
Reviewer: 1) Reviewers required for all obtained compounds, in addition to 1H NMR spectroscopy data, elemental analysis or high-resolution mass spectrometry to confirm their composition and purity, instead Authors added TOF MS (ES+) data only for some compounds and 13C NMR spectra for 2 final compounds. In addition, to justify that 1H NMR spectra are not reported in the supplementary material, Authors answered “Because some compounds have poor solubility, 1H-NMR spectra can't be tested again, so they are not provided”. This is not acceptable. Authors must to report, at least for ALL final compounds, NMR spectra (to confirm the structure) and HRMS or elemental analysis (to confirm purity).
Response: Thank you for your attention and careful review of our manuscript. According to your suggestion, we have completed the 1H NMR and TOF MS (ES+) data of all target compounds in the revised supplementary materials. At the same time, the TOF MS (ES+) data were marked in red in the corresponding position in the revised manuscript. For the 13C NMR data of the target compounds, we did not provide it due to the problem of the amount and solubility of the target compound, we are very sorry about this.
2) In the Experimental Section, 3.2. Chemistry. If intermediates are newly synthesized compounds, Authors have to provide sufficient information for chemical characterization, at least 1H NMR spectra to confirm the structure and HRMS or elemental analysis to confirm purity. Alternatively, if compound were previously synthesized, it is not needed to describe they again, and the related references have to be inserted. The title of all paragraphs has to report the correct nomenclature, not only “General Procedure for the Preparation of Compounds …”
Response: Thanks to the reviewer’s advice and it is useful to improve our paper. We have added the 1H NMR and TOF MS (ES+) data of the intermediates in the supplementary materials. However, I'm sorry that I cannot provide intermediates 17a-b and 18a-b 1H NMR data, because the shape of both intermediates were oily and difficult to dissolve and treat. Our approach to these two steps was to directly dissolve them with a solvent and put them in the next step of the reaction. Then they were purified by column chromatography in the final post-treatment. Of course, we also tried to dissolve them with DMSO and test their 1H NMR data, but this caused the spectrometer jam. Therefore, we were unable to provide 1H NMR data of these two intermediates 17a-b and 18a-b, and we are sorry again. Although we were unable to provide 1H NMR data on intermediates 17a-b and 18a-b, we performed TOF MS (ES+) tests on intermediates 17a-b and 18a-b and determined their structure and purity.
Reviewers' comments: Minor concerns. We noted numerous vagueness through all the manuscript. Here is a list of representative ones. Please, check carefully all the manuscript to correct formal and inattention mistakes.
1) Line 33: “The third-generation EGFR inhibitor of Olmutinib”, delete “of”.
Response: Thanks for your kindly reminding. I am sorry, it is my carelessness. The corresponding item was revised in the revised manuscript.
2) Line 52: change “retain” with “retained”.
Response: Thanks for your kindly reminding. I am sorry, it is my carelessness. The corresponding item was revised in the revised manuscript.
3) Lines 52-55 “For the second series of compounds (Figure 4), we retain the thiophene and pyrimidine core, and introduced the triazine structure of Phosphatidylinositol-3-kinaset (PI3K) inhibitor ZSTK474 (5) at the same time, the structures of thiophene and pyrimidine were transformed into triazine and pyrimidine” It’s not clear to which compounds you are referring to.
Response: Thanks for your kindly reminding. I am sorry that it is not clearly stated in the manuscript, it is my carelessness. The compound here referred to the second series of target compounds. The corresponding item was revised in the revised manuscript.
4) Lines 108-110: “Since the side chains of the benzene ring are conjugated, anti-proliferative activities of compounds H1-H6 that were substituted with cyano groups showed moderate inhibitory activity against all the cell lines.” What does mean this sentence?
Response: Thanks for your kindly reminding. I am sorry, I'm making a mistake here. In the manuscript, I have revised to " After the introduction of electron-withdrawing cyanide, anti-proliferative activities of compounds H1-H6 that were substituted with cyano groups showed moderate inhibitory activity against all the cell lines." The corresponding item was revised in the revised manuscript.
5) Line 80: “The synthetic routes of target compounds H1-H8 are outlined in Schemes 1-2.” This is not true. The synthesis of compounds H1-H8 is described only in the scheme 1.
Response: Thanks for your kindly reminding. I am sorry, I'm making a mistake here. In the manuscript, I have revised to " The synthetic routes of target compounds H1-H16 are outlined in Schemes 1-2." The corresponding item was revised in the revised manuscript.
6) Line 83: “thiophen-2-ylboronic” change in “thiophen-2-yl-boronic”.
Response: Thanks for your kindly reminding. I am sorry, it is my carelessness. The corresponding item was revised in the revised manuscript.
7) Line 92: “As shown in Scheme 1, We used commercially available 2,4,6-Trichloropyrimidine” change the capital letter in “We” and in “Trichloropyrimidine”.
Response: Thanks for your kindly reminding. I am sorry, it is my carelessness. The corresponding item was revised in the revised manuscript.
8) Line 138: “than the control drug of Olmutinib” delete “of”.
Response: Thanks for your kindly reminding. I am sorry, it is my carelessness. The corresponding item was revised in the revised manuscript.
9) Line 151: figure 3 is wrong. More probably figure 5 is correct. Please check all times that figure 3 is reported instead of figure 5.
Response: Thanks for your kindly reminding. I am sorry, I wrote Figure 5 as Figure 3, it is my carelessness. The corresponding item was revised in the revised manuscript.
10) Line 174: change “1H-NMR” with 1H-NMR.
Response: Thanks for your kindly reminding. I am sorry, it is my carelessness. The corresponding item was revised in the revised manuscript.

Reviewer 2 Report

Thanks to the authors for improving the manuscript and answering my questions and comments.

Author Response

(The authors gave the same response as above.)
